# The Ameliorative Effects of Fucoidan in Thioacetaide-Induced Liver Injury in Mice

**DOI:** 10.3390/molecules26071937

**Published:** 2021-03-30

**Authors:** Ming-Yang Tsai, Wei-Cheng Yang, Chuen-Fu Lin, Chao-Min Wang, Hsien-Yueh Liu, Chen-Si Lin, Jen-Wei Lin, Wei-Li Lin, Tzu-Chun Lin, Pei-Shan Fan, Kuo-Hsiang Hung, Yu-Wen Lu, Geng-Ruei Chang

**Affiliations:** 1Animal Industry Division, Livestock Research Institute, Council of Agriculture, Executive Yuan, 112 Muchang, Xinhua Dist, Tainan 71246, Taiwan; mytsai@mail.tlri.gov.tw; 2Graduate Institute of Bioresources, National Pingtung University of Science and Technology, 1 Shuefu Road, Neipu, Pingtung 91201, Taiwan; 3School of Veterinary Medicine, National Taiwan University, 4 Section, 1 Roosevelt Road, Taipei 10617, Taiwan; yangweicheng@ntu.edu.tw (W.-C.Y.); cslin100@ntu.edu.tw (C.-S.L.); 4Department of Veterinary Medicine, College of Veterinary Medicine, National Pingtung University of Science and Technology, Shuefu Road, Neipu, Pingtung 912301, Taiwan; cflin2283@mail.npust.edu.tw; 5Department of Veterinary Medicine, National Chiayi University, 580 Xinmin Road, Chiayi 60054, Taiwan; leowang@mail.ncyu.edu.tw (C.-M.W.); lin890090@gmail.com (T.-C.L.); babybelle349@gmail.com (P.-S.F.); 6Bachelor Degree Program in Animal Healthcare, Hungkuang University, 6 Section, 1018 Taiwan Boulevard, Shalu District, Taichung 433304, Taiwan; lhy_vet@hk.edu.tw (H.-Y.L.); jenweilin@hk.edu.tw (J.-W.L.); ivorylily99@gmail.com (W.-L.L.); 7General Education Center, Chaoyang University of Technology, 168 Jifeng Eastern Road, Taichung 413310, Taiwan; 8Department of Chinese Medicine, Show Chwan Memorial Hospital, 1 Section, 542 Chung-Shan Road, Changhua 50008, Taiwan; 9Department of Chinese Medicine, Chang Bing Show Chwan Memorial Hospital, 6 Lugong Road, Changhua 50544, Taiwan

**Keywords:** fucoidan, inflammation, liver, mice, thioacetamide

## Abstract

Liver disorders have been recognized as one major health concern. Fucoidan, a sulfated polysaccharide extracted from the brown seaweed Fucus serratus, has previously been reported as an anti-inflammatory and antioxidant. However, the discovery and validation of its hepatoprotective properties and elucidation of its mechanisms of action are still unknown. The objective of the current study was to investigate the effect and possible modes of action of a treatment of fucoidan against thioacetamide (TAA)-induced liver injury in male C57BL/6 mice by serum biochemical and histological analyses. The mouse model for liver damage was developed by the administration of TAA thrice a week for six weeks. The mice with TAA-induced liver injury were orally administered fucoidan once a day for 42 days. The treated mice showed significantly higher body weights; food intakes; hepatic antioxidative enzymes (catalase, glutathione peroxidase (GPx), and superoxide dismutase (SOD)); and a lower serum alanine aminotransferase (ALT), aspartate aminotransferase (AST), tumor necrosis factor-α (TNF-α), interleukin-1β (IL-1β), and C-reactive protein (CRP) levels. Additionally, a reduced hepatic IL-6 level and a decreased expression of inflammatory-related genes, such as cyclooxygenase-2 (COX-2), and inducible nitric oxide synthase (iNOS) mRNA was observed. These results demonstrated that fucoidan had a hepatoprotective effect on liver injury through the suppression of the inflammatory responses and acting as an antioxidant. In addition, here, we validated the use of fucoidan against liver disorders with supporting molecular data.

## 1. Introduction

Liver is a vital organ that coordinates several major functions to maintain the homeostasis of the body. It plays the crucial role in the metabolism of sugars, proteins, lipids, vitamins, and hormones. The liver is also an important detoxification organ. When substances are absorbed into the body, blood can flow directly from the surface of the small intestine to the hepatic portal vein. Since the liver’s cell membrane presents an optimal permeability and contacts directly with the blood [1], toxic substances brought by the hepatic portal vein can be eliminated after being metabolized by the liver [2]. In addition, inflammation and wound healing are interrelated processes, since inflammatory signals stimulate immune cells toward injury sites [3]. The repair and regeneration of the injured tissues is caused by apoptosis and regenerative cellular processes [4]. Additionally, the accumulation extracellular matrix and scarring tissues are the endpoints of inflammation. However, in the case of chronic injury, the wound-healing process becomes maladaptive, causing the loss of functional hepatic parenchyma. This can further develop into liver fibrosis, which may be a percussor for hepatic cirrhosis and liver cancer [5,6]. Possible causes of abnormal liver function include viral infections, alcohol intake, autoimmunity, genetics, hepatobiliary tumors or infections, chemical toxins, and drugs. Around 50% of liver damage is caused by drugs [7].

Hepatotoxic molecules have the ability to react with cell components and induce different types of lesions in the liver. The chemical toxins such as acetaminophen, carbon tetrachloride (CCl4), galactosamine, and thioacetamide have been efficiently used in the past in vitro, as well as in vivo, to create models of experimental hepatocyte injury [8,9]. The process of hepatotoxicity caused by thioacetamide (TAA) is recognized to be similar to that in humans [10]. TAA-induced hepatic inflammation, fibrosis, and liver damage in mice, in terms of clinical features, are very close to chronic human liver disease [11]. After TAA exposure, the levels of plasma alanine aminotransferase (ALT) and aspartate aminotransferase (AST) are upregulated due to hepatocyte membrane damage, which causes a leakage of transaminase enzymes into the systemic circulation [12]. The advantage of using the TAA model for the hepatotoxin animal model is its unique specificity for targeting the liver [12,13]. The toxicity is because of its bioactivation led by a mixed-function oxidase system—mainly, the cytochrome P2E1 and flavin adenine dinucleotide monooxygenases [13,14]. Furthermore, the activation of TAA causes the formation of reactive metabolites such as thioacetamide-S-oxide radicals and reactive oxygen species (ROS) intermediates [13]. Hence, the biochemical processing of TAA within the cellular milieu causes liver oxidative damage, which can further degenerate and cause necrosis in liver tissues [15].

Fucoidan belongs to a class of fucose-rich sulfated carbohydrates usually present in a brown, marine algae, echinoderms [16], and seagrasses [17]. Fucoidan-rich brown algae is commercially available as a dietary supplement or nutraceutical. With the advantages of having low toxicity and high oral bioavailability, it has been studied for its potential pharmacological effects. Pharmacologically, fucoidan mediates the pathogenesis of numerous cellular processes, such as inflammation, carcinogenesis, and oxidative damage [18,19]. Fucoidan has also been reported to cause cytotoxicity and induce apoptosis in various cancer cells [20], including MCF-7 breast cancer cells, in vitro [21] and in Lewis lung adenocarcinoma cells transplanted into mice [22]. In addition, pronounced antitumor and antimetastatic effects of fucoidan were observed in mice with transplanted lung adenocarcinoma [23]. T-cell-mediated natural killer (NK) cells were also efficiently upregulated in mice treated with fucoidan, and NK cell activation was found to be associated with an increased production of interferon (IFN)-γ and interleukin (IL)-12 by splenic T cells [24]. Fucoidan has also been observed to suppress angiogenic activity while downregulating the vascular endothelial growth factor (VEGF) receptor expression and inhibiting the proliferation of VEGF-induced human umbilical vein endothelial cells [25]. Additional studies have reported the inhibitory activity of fucoidan against matrix metalloproteinases and nuclear factor-kappa β (NF-κβ), preventing metastasis in tumor mice models [26]. Overall, fucoidan is considered to provide anticancer immunity via the activation of an immune cell influx and by upregulating the production of anticancer cytokines.

In other reports, concanavalin A-induced liver injury and the concomitant increase of plasma tumor necrosis factor-α (TNF-α) and IFN-γ levels were inhibited by fucoidan; however, the plasma IL-10 levels were enhanced [27]. It has also been proven that fucoidan has a protective effect against liver ischemia injury via suppression of the inflammatory activation pathway, several inflammatory mediators, and cell infiltration in the inflamed sites [28]. The mechanisms of action of fucoidan against liver injury caused by TAA in male C57BL/6J mice are not completely understood. Therefore, in this study, we analyzed the hepatic function levels, inflammation cytokines, histopathology, and antioxidative enzymes in mice with liver injuries undergoing treatment with or without fucoidan to better understand the mode of action of fucoidan.

## 2. Results

### 2.1. Fucoidan Affected Body Weight and Food Intake in Mice with TAA-Induced Liver Damage

Significant differences in body weight, daily food intake, daily body weight gain, and daily food efficiency were observed between the control (saline-treated TAA-induced liver injury mice) and fucoidan group (fucoidan-treated TAA-induced liver injury mice) (Figure 1). TAA was fed to male C57BL6/J mice for eight weeks to induce hepatotoxicity. After six weeks of fucoidan treatment, the fucoidan-treated mice exhibited greater body weights, food intakes, weekly body weight gains, and daily food efficiency than the controls. The body weights and food intakes after six weeks, for all experimental groups, are represented in Figure 1a,b, respectively. Mice with TAA-induced liver damage treated with fucoidan presented 1.07-fold higher body weights than the control mice; this difference was significant (Figure 1a). Moreover, in comparison to the control mice, it was observed that the food intake of the treated mice increased by 1.28-fold (Figure 1b). The fucoidan-treated mice presented significantly higher weekly body weight gains by 1.61-fold (Figure 1c) than that of the control mice. In addition, these changes were demonstrated, along with the significant changes in the daily food efficiency in the group treated with curcumin, as 1.27-fold higher than the control mice (Figure 1d). However, the body weight, daily food intake, daily gain in body weight, and daily efficiency in food intake were all significantly lower in the fucoidan-treated group than in the control group.

### 2.2. Fucoidan Affected Serum ALT, AST, and Liver Weight in Mice with TAA-Induced Liver Damage

There were significant differences in the serum ALT, AST, and liver weights among the three groups. In fucoidan-treated mice, the serum ALT and AST were downregulated by 23% (Figure 2a) and by 45% (Figure 2b), respectively. This represents a significant downregulation of the hepatic function markers when in comparison to the TAA-induced liver injury control mice. In addition, the fucoidan-treated group exhibited a 27% elevation in liver weights (Figure 2c) and a 21% elevation in body weight-normalized liver weights (Figure 2d). In the normal group, the serum ALT and AST, liver weights, and the relative liver/body weight ratios were downregulated when compared to those in fucoidan-treated mice.

### 2.3. Fucoidan Affected the Serum Alkaline Phosphatase (ALP), Bilirubin, Globulin, and γ-Glutamyl Transferase (γ-GT) 

Additionally, we also determined the effects of fucoidan on the serum enzymes, ALP, bilirubin, globulin, and γ-GT levels in the TAA-induced hepatotoxicity mice models. The results are shown in Figure 3. The levels of ALP, bilirubin, globulin, and γ-GT of the TAA-treated group after an intraperitoneal (ip) injection dose of 100 mg/kg were significantly elevated after six weeks of treatment when compared to the control mice. The increased levels of these biochemical biomarkers match with the hepatic cell damage [29]. In the fucoidan-treated mice, the levels of serum ALP (15%), bilirubin (12%), globulin (21%), and γ-GT (20%) were significantly decreased, demonstrating the overall successful optimization of the hepatic injuries. Overall, TAA mice treated with fucoidan showed a significant downregulation of ALT, AST, and ALP in the serum, confirming the role of fucoidan in the recovery of the liver function. However, these biomarker levels in fucoidan-treated mice did not decrease to the normal baseline observed in healthy mice.

### 2.4. Fucoidan Affected Liver Damage 

The histopathological changes, observed with hematoxylin and eosin (H&E) staining for all the mice groups, are shown in Figure 4. In the histological profile, the normal control group presented regular hepatocytes with well-preserved cytoplasms, prominent nuclei, and nucleoles. No sign of inflammation, fatty change, or necrosis was noted. However, in TAA-treated mice, the liver tissue showed extensive injuries, with a multifocal area of the necrosis of hepatocytes, cell swelling, disruption of the membranes, and contraction of the nucleus. Fucoidan-treated mice presented markedly diminished histological alterations (Figure 4a). The regular hepatocyte structure was observed with much reduced ballooning and tissue degeneration. The necrotic activity was also decreased in comparison to the TAA-treated group. Additionally, the liver damage score, using Suzuki scoring (0–4), decreased markedly for fucoidan-treated mice (Figure 4b) [30,31].

### 2.5. Fucoidan Affected Tumor Necrosis Factor-ɑ (TNF-ɑ), Interleukin-1β (IL-1β), Fibroblast Growth Factor-21 (FGF21), C-Reactive Protein (CRP), and Cytokine Levels in the Serum 

The levels of serum inflammatory cytokines were shown to be significantly altered among the three groups analyzed (Figure 5). The serum TNF-ɑ levels were 18% lower in the fucoidan treatment group than that of the TAA-induced liver damage group (Figure 5a). Compared to the TAA group, the serum IL-1β levels were 39% lower in the fucoidan-treated TAA-induced liver damage group (Figure 5b). Additionally, we also analyzed serum FGF21, a regulator of hepatic metabolic pathways, to improve the steatosis and inflammation [32]. The serum FGF21 levels in fucoidan-treated-mice were 1.18-fold higher than in TAA-induced liver-damaged mice (Figure 5c). In contrast, the serum CRP levels of fucoidan-treated mice were 25% lower than in the mice with TAA-induced liver damage (Figure 5d). Compared to normal mice, the liver damage model treated with fucoidan revealed a 1.25-fold upregulation in the TNF-ɑ levels, 1.50-fold upregulation in IL-1β, and 47% decrease in FGF21, with a 1.87-fold increase in the serum CRP levels.

### 2.6. Fucoidan Affected Interleukin-6 (IL-6), Patatin-Like Phospholipid Domain Containing Protein 3 (PNPLA3), the mRNA of Liver Fatty Acid-Binding Protein (L-FABP), Cyclooxygenase-2 (COX-2), and Inducible Nitric Oxide Synthase (iNOS)

IL-6 is known to induce hepatic inflammatory cell infiltration [33], while PNPLA3 represents a modifier of progression of hepatocellular injury and liver-associated disorders such as steatohepatitis, chronic hepatitis, and hepatocellular carcinoma by promoting the release of inflammatory cytokines [34,35]. Here, we found that the expression of IL-6 and PNPLA3 were significantly different between the three analyzed groups by using a Western blotting analysis (Figure 6a). The expression of hepatic IL-6 was 39% lower in fucoidan-treated mice than the mice with TAA-induced liver injury (Figure 6b). Compared to the TAA group, the hepatic PNPLA3 expression was 36% lower in fucoidan-treated mice (Figure 6c). Contrarily, the mRNA levels of L-FABP in the liver were 31% lower in the latter group (Figure 6d). Moreover, the hepatic mRNA levels of COX-2 (Figure 6e) and iNOS (Figure 6f) were 29% and 17% lower, respectively, in mice under fucoidan treatment than that of mice with TAA-induced liver damage.

In contrast, the hepatic IL-6 and PNPLA3 expression of fucoidan-treated TAA-induced liver-damaged mice were 1.85-fold and 3.78-fold higher, respectively, compared to normal mice treated with saline. Additionally, the hepatic L-FABP, COX-2, and iNOS mRNA levels of mice treated with fucoidan in TAA-induced liver damage revealed increases of 1.69-fold, 4.58-fold, and 2.23-fold, respectively, when compared to normal mice.

### 2.7. Fucoidan Affected Antioxidant Enzymes and ROS in the Liver

We next evaluated whether fucoidan improved liver damage, as the evidence indicated from the TAA treatment results in the development of liver injury by inducing oxidative damage [13,14]. A significant difference in the liver antioxidant enzymes, such as superoxide dismutase (SOD), catalase, and glutathione peroxidase (GPx), was observed in our study (Figure 7). The progress of a liver injury is closely related to the downregulation of antioxidant enzymes in the liver; when the activity of these enzymes is increased, the liver function is enhanced, counteracting the hepatotoxicity issues [36]. Indeed, mice with TAA-induced liver damage treated with fucoidan exhibited significant increases of 1.59-fold, 4.12-fold, and 1.41-fold of catalase (Figure 7a), GPx (Figure 7b), and SOD (Figure 7c), respectively. The overproduction of hepatic ROS was predominantly accelerated in liver inflammatory injuries, affecting the hepatic structure and subsequently leading to severe hepatocellular dysfunction with a poor prognosis [37,38]. Here, we also found that fucoidan-treated mice expressed a significant decrease in hepatic ROS by 29% compared to the liver damage control mice (Figure 7d). The increase in hepatic antioxidant enzymes was lowered by 21%, 22%, and 18% for catalase, GPx, and SOD, respectively, in fucoidan-treated liver-damaged mice compared to normal mice. In contrast, the hepatic ROS production was increased by 1.20-fold in fucoidan-treated mice.

## 3. Discussion

Liver diseases, whether caused by viral infections or drugs, are considered severe health issues and require immediate treatment with minimal toxic side effects [39]. Hepatic inflammation is common to all forms of liver injury, including cirrhosis, fibrosis, and liver cancer. Therefore, beneficial anti-inflammatory molecules may be considered as potential therapeutic agents for liver diseases [40]. Natural products present fewer side effects and are often associated with greater biocompatibility [41]. The main impediments to employing natural products involve establishing a comprehensive knowledge of medicinal plants, purifying their components, and identifying their mechanisms of action. In the current study, we successfully established a TAA-induced hepatic injury model to determine the benefits of fucoidan, a plant-based medicinal bioactive compound. In the TAA and fucoidan group, mice were administered fucoidan through oral gavage once per day for 42 days. We then measured the body weight; food intake; and levels of liver function enzymes, antioxidant enzymes, inflammatory cytokines, and molecular proteins of the mice and performed a histological examination, RNA extraction, and quantitative polymerase chain reaction (PCR) analysis. Finally, we demonstrated that fucoidan induced the regulation of the hepatic inflammatory and related regulatory pathways.

Following a TAA treatment for six weeks, this molecule contributed significantly to the decrease in body weight and body weight gain when compared to the control mice. A decrease in weight gain occurs when the food intake decreases as a result of a decrease in appetite [42]. This supports the idea that rats with TAA-induced liver injuries undergo changes in the brain’s tryptophan, resulting in a decrease in food intake associated with anorexia [43]. Fucoidan can retard body weight loss, change anorexia, and enhance the quality of life in cancer patients under chemotherapy [44]. Additionally, the expression of insulin-like growth factor 1 (IGF-1) and the mammalian target of rapamycin (mTOR)/S6 ribosomal protein kinase 1 protein are elevated when under the effect of chemotherapeutic drugs such as cisplatin and gemcitabine [45]. Moreover, reports have indicated that the IGF-1 and mTOR pathways are linked to an increased caloric intake and elevate their expression toward an enhanced growth/weight [46,47]. Taken together, fucoidan is known to act as a potential agent to improve liver damage-related anorexia.

The ALT and AST levels in the serum are the vital indicators of liver damage [41]. The treatment of TAA causes major changes in the histology of liver and, also, significantly upregulates the ALT and AST levels in the serum [48]. Fucoidan ameliorated the damage and attenuated the liver histological damage with the significant downregulation of the ALT and AST levels in comparison to the TAA model group. On the other hand, the serum levels of liver enzymes such as ALP, bilirubin, globulin, and γ-GT increase [29]. These enzymes are biomarkers for liver injury. In our findings, considerable elevations in ALT, AST, ALP, the total proteins, and globulin in the serum were also observed [41,49]. This was in line with previous findings. Mohamed et al. showed that fucoidan ameliorated diazinon-induced injuries to hepatic tissues by changing the outflow of hepatic transaminases to the bloodstream [50]. Additionally, increased serum globulin resulted from inflammation, infection, tissue necrosis, and stress [51]. Here, TAA-induced liver damage treated with fucoidan could decrease the serum globulin levels, which was attributed to fucoidan improving the hepatic inflammatory response in high-fat high-sucrose diet-fed mice [52]. In addition, after the treatment with fucoidan, a significant recovery of hepatic damage was observed by the downregulated plasma levels of the hepatic enzymes. These results validated the hepatoprotective activity of fucoidan.

Apart from the biochemistry, similar effects were validated from our histopathological results. Overall, 100-mg/kg TAA applied thrice a week successfully induced chronic liver damage in mice [49,53], while mice treated with TAA had lower body weights compared with the untreated mice [54]. The lower body weights in the TAA-treated mice were different from most nonalcoholic fatty liver disease models, since the liver damage was not caused by dieting or factors related to obesity [42,47]. Additionally, some studies have indicated that animals treated with TAA for over eight weeks have higher liver weights due to the accumulation of collagen and extracellular matrix protein in the liver after TAA exposure [54,55]. We found, using Sirius red staining, that a liver injury induced with a treatment of TAA for six weeks did not develop liver fibrosis (Appendix A). The discrepancies of our results with other studies are probably indicative of the effects of the length of the TAA treatment. Whether liver damage affects fibrogenesis, many studies have suggested a hepatoprotective role of medicinal plants in TAA injury models [53]. In the present animal model, the fucoidan treatment offered protection to mice for substantial lethality and, also, because it was able to lower the ALT, AST, and ALP levels in the serum. After 42 days of fucoidan treatment, mice markedly improved their histological changes, with less degeneration and necrosis. In chronic liver injury experiments done in combination with other studies, fucoidan was shown to promote the restoration of hepatic physiology, further confirmed with the ALT, AST, ALP, bilirubin, globulin, and γ-GT levels [56].

Inflammation is a common stimulus for all liver diseases, triggering the progression of liver damage to severe fibrogenesis or culminating in hepatic carcinoma [57]. Therefore, we investigated the proteins such as TNF-α, IL-1β, FGF21, and CRP that have an effect on the inflammatory response. TNF-α is a pleiotropic proinflammatory cytokine that can induce multiple mechanisms to initiate apoptosis in hepatocytes and, hence, cause liver injury by either directly or indirectly targeting the mitochondria [58]. The research indicates that the TNF receptor one knockout mice showed reduced CCl4-induced liver fibrosis [59]. There is no doubt that TNF-α can promote fibrosis and lead to chronic liver injury and inflammation. Interleukin-1β is associated with the activation of NF-κβ signaling, the upregulation of proinflammatory cytokines, and liver damage [60]. Another study showed that mice with an IL-1β deficiency had less diet-induced inflammation and liver fibrosis. In addition, a decreased expression of IL-6, TNF-α, and TGF-β, in comparison to wild-type mice, was observed [61]. Next, the elevated FGF21 was able to lower the content of AST, ALT, and γ-GT [62]. For example, acetaminophen-induced hepatotoxicity and mortality was exacerbated in FGF21 null mice. The opposite effect was observed when the treatment with recombinant FGF21 produced significant protective effects on the overall liver function [63]. Finally, the CRP levels correlated closely with the changes in inflammation/disease activity, radiological damage, and functional disability [64]. CRP is primarily synthesized in the liver and is involved in many chronic diseases. Moreover, the elevated CRP level reveals that chronic ethanol administration is responsible for severe hepatic endothelial injury [65]. Our results demonstrated that TNF-α, IL-1β, and CRP in fucoidan-treated mice were lower than of TAA-treated mice. On the other hand, FGF21 was enhanced after the treatment with fucoidan. This means that fucoidan is able to reduce the inflammation and severity of liver damage. In light of all our findings, fucoidan demonstrated a significant hepatoprotective effect against TAA-induced liver injuries by regulating the inflammatory process.

The other important factors related to liver damage are IL-6, PNPLA3, L-FABP, COX-2, and iNOS. IL-6, upregulated in chronic liver inflammation [66], is produced by activated Kupffer cells exacerbating the local inflammatory response and inducing hepatocyte proliferation, with a compensatory transformation of hepatocytes into malignant hepatocytes [67]. PNPLA3 increases the release of proinflammatory and profibrogenic cytokines, known to be active key players in the liver injury process [68,69]. Human hepatic stellate cells with the PNPLA3 variant have a higher expression of inflammatory cytokines and chemokines [70]. L-FABP can be used for the diagnosis of acute hepatitis, chronic hepatitis, and cirrhosis [71]. In another study, it was demonstrated that the upregulated serum L-FABP levels are related to a degree of fibrosis and inflammation in the liver, which indicates that serum L-FABP can be a noninvasive biomarker used to assess the severity of fibrosis and inflammation in nonalcoholic steatohepatitis patients [72]. The COX-2 expression was related to the early phases of different chronic liver diseases and found to be responsible for the induction of hepatic cancer [73]. A real-time quantitative PCR analysis of total RNA demonstrated that the hepatic COX-2 gene expression is upregulated in both acutely and chronically damaged livers [74]. Finally, iNOS has been also reported to be involved in the excessive production of proinflammatory mediators [75]. Chronic liver diseases begin with an inflammatory phase, followed by fibrosis and continuous oxidative stress. In this state, the iNOS production is increased, causing an overproduction of nitric oxide [76]. By using Western blotting, we can find that the expression of IL-6 and PNPLA3 are reduced by fucoidan. Moreover, the gene expression of L-FABP, COX-2, and iNOS are also lower in fucoidan-treated mice than in TAA-treated mice. The application of fucoidan was able to reduce the inflammation and liver damage-related factors, reducing the effect of the liver damage caused by TAA.

Oxidative stress is a major factor in the progression of liver damage and its pathogenesis. ROS cause the induction of hepatic inflammation, necrosis, and cholestasis [77]. Additionally, the antioxidant enzymes, including catalase, GPx, and SOD, offer protection against the harmful effects of ROS [78]. GPx is a family of enzymes that constitutes a main antioxidant defense system in mammals [79]. The GPx and SOD levels are regularly downregulated, with an accompanied increase in the ALT and AST levels [80]. CAT is present in peroxisomes, where it decomposes two hydrogen peroxide (H_2_O_2_) molecules into two H_2_O molecules and O_2_ [81]. While the concentration of H_2_O_2_ increases, CAT shows a greater contribution to H_2_O_2_ degradation [82]. The serum CAT activity was moderately increased in fatty liver and acute alcoholic hepatitis [83]. Our results indicate that fucoidan decreases oxidative stress and the contents of ROS by increasing the GPx, SOD, and CAT activity. Therefore, the treatment with fucoidan was able to reduce the ROS to prevent liver cells from developing into a more severe liver disease.

Together, our findings demonstrated and validated that fucoidan showed hepatoprotective activity that was mediated via the inhibition of the inflammatory pathways and upregulation of the antioxidant enzymes. Hence, we emphasized the importance of fucoidan as a new potential treatment against liver damage.

## 4. Materials and Methods

### 4.1. Animal, Liver Injury Mouse, and Fucoidan

Male C57BL/6J mice, five-weeks-old, 18–20 g, were purchased from Education Research Resource, National Laboratory Animal Center, Taipei, Taiwan. In accordance with the Taiwan government’s recommendations, all animal housing and experimentation were performed under the Guidelines for the Care and Use of Laboratory Animals. The review of our experimental protocol was conducted by the National Chiayi University’s Institutional Animal Care and Use Committee, who approved it under the archive no. 108017. Animals were housed in normal cages at 22 ± 1 °C under 55% ± 5% humidity, with a 12:12-h light–dark schedule. The mice were fed a standard diet (SD, 3.3 kcal/g of metabolizable energy; diet 5008; PMI Nutrition International, Brentwood, MO, USA) for 4 weeks ad libitum until the mice weighed approximately 25 g. They were subsequently divided into three groups.

Mice fed the SD were randomly divided into the normal control, TAA model, and TAA and fucoidan groups. Within each housing group, the mice were randomly assigned to three groups consisting of 10 mice. The mice in the TAA model and fucoidan groups were injected with 100-mg/kg TAA ip (Sigma, St. Louis, MO, USA) thrice a week for six weeks, and the mice in the fucoidan group were administered fucoidan from *Fucus vesiculosus* (20 mg/kg; Sigma) by oral gavage once a day for 42 days. The dose of TAA was based on the available literature [49,53,54], and the dose of fucoidan was based on previous articles that revealed fucoidan as a choice candidate for chronic treatment in blood glucose homeostasis, lung injury, hepatic ischemia–reperfusion injury, and endometriotic studies in mouse models [84,85,86]. In our preliminary studies, the serum ALT, AST, globulin, and γ-GT levels in the fucoidan-treated (10 mg/kg) TAA-induced liver injury mice did not differ significantly from those of the TAA-induced liver injury mice (Appendix A). Consequently, we initiated this study by using a dose of fucoidan at 20 mg/kg in our mice model. All mice were weighed weekly. On termination of our experiment, mice were euthanized and their livers harvested, as well as their serum, for the subsequent analysis of the liver function enzymes, antioxidant enzymes, histological examination, inflammatory cytokine, molecular proteins, RNA extraction, and quantitative PCR analysis.

### 4.2. Body Weight and Food Intake

Weekly measurements of the food intake, as well as body weight, throughout the study period were taken. For the food intake measurements, the leftover food within each cage dispenser, in addition to what was spilled, was measured.

### 4.3. Liver Function Enzymes Tests

The levels of ALT, AST, ALP, bilirubin, globulin, and γ-GT in the serum were measured from the blood samples using an automated chemistry analyzer (Catalyst One Chemistry Analyzer, IDEXX Laboratories, Westbrook, ME, USA) from a commercial kit. Our method followed the manufacturer’s recommendations. The coefficient of variation within and between the analysis runs was <2%.

### 4.4. Liver Histopathological Evaluation

We measured the liver weights and determined their percentages in relation to the total body weight. The isolated liver tissues were treated with formalin, entrenched in paraffin, and further processed for the histological analysis. The sections (3 μm) were stained with H&E staining, and the liver damage was quantified. The injuries were classified using the mentioned four-point scale (Suzuki scoring 0–4). Scoring was assigned based on the degree of liver damage: 0 = normal, 1 = the development of sinusoidal congestion space, 2 or 3 = the presence and/or severity of sinusoidal congestion and cytoplasmic vacuolization, and 4 = the necrosis of parenchymal cells and hemorrhage [30,31].

All the sections were reassessed for liver damage changes by Taiwan-certified veterinary pathologists blinded to the animals’ treatments.

### 4.5. Determination of Serum Inflammatory Cytokines, FGF21, and CRP

The levels of TNF-ɑ and IL-1β in the plasma were determined using mouse TNF-α and IL-1β ELISA kits (Invitrogen, Carlsbad, CA, USA) according to the manufacturer’s instructions. Serum mouse FGF-21 ELISA kits were obtained from Zgenebio Biotech (Taipei, Taiwan). Serum concentrations of CRP were measured using a mouse CRP quantitative ELISA kit (Abcam, Cambridge, UK).

### 4.6. Western Blotting Assay

Animals were anesthetized with an ip injection at the termination of the experiment. Livers were quickly removed, minced coarsely, and immediately homogenized prior to the Western blot [47,49] using the following antibodies: anti-IL-6 (Cell Signaling Technology, Beverly, MA, USA), anti-PNPLA3 (Sigma), and anti-actin antibodies (Sigma). We used enhanced chemiluminescence reagents (Thermo Scientific, Rockford, MA, USA) to produce immunoreactive signals and UVP ChemStudio (Analytik Jena, Upland, CA, USA) to detect these signals. Protein expression and phosphorylation studies were quantified using ImageJ by the National Institutes of Health (Bethesda, MA, USA).

### 4.7. RNA Extraction and Real-Time Quantitative PCR

The mRNA levels of L-FABP, COX-2, and iNOS in the livers were determined by the CFX Connect Quantitative Real-Time PCR System (Bio-Rad, Hercules, CA, USA). Briefly, the total RNA was isolated from the liver samples using TRI Reagent (Sigma), and the concentration was assessed based on the absorbance at 260–280 and 230–260 nm on a Qubit fluorometer (Invitrogen, Carlsbad, CA, USA). Next, we performed real-time PCR using this complementary DNA and iTaq universal SYBR Green supermix (Bio-Rad), in accordance with the manufacturer’s protocols. The PCR was performed as follows: 95 °C for 5 min and then 45 cycles at 95 °C for 15 s, followed by 60 °C for 25 s. The L-FABP sequence primers employed in this study were forward 5′-GCAGAGCCAGGAGAACTTTG-3′ and reverse 5′-GGGTCCATAGGTGATGGTGAG-3′ [87]; those for COX-2 were forward 5′- AAAACCGTGGGGAATGTATGAGC-3′ and reverse 5′-GATGGGTGAAGTGCTGGGGAAAG-3′ [88]; those for the iNOS sequence primers employed in this study were forward 5′- TCTTGGGTCTCCGCTTCTCGTC-3′ and reverse 5′-TGGCTGGTACATGGGCAC AGAG-3′ [89]. Each target gene expression level was calculated in relation to the Actb level and expressed using the 2−ΔΔCt method.

### 4.8. Evaluation of Hepatic Catalase, GPx, SOD, and ROS

An evaluation of the functional activity of the antioxidant system was analyzed for the enzymatic activity of hepatic catalase, GPx, and SOD. Livers were perfused with ice-cold saline (0.9% sodium chloride) and homogenized in chilled potassium chloride (1.17%) [90]. After, the obtained homogenates underwent centrifugation at 800× *g* for 5 min at 4 °C. The supernatant was recentrifuged at 10,500× *g* for 20 min at 4 °C to finally obtain the post-mitochondrial supernatant. This was used for measuring the catalase, GPx, and SOD. The enzymes were quantified using a commercially available colorimetrical kit (#K773-100 for catalase, #K762-100 for GPx, and #K335-100 for SOD; BioVision, Milpitas, CA, USA). In addition, small portions of fresh liver tissue were collected and homogenized in an ice-cold lysis solution (10-mM Tris, 150-mM NaCl, 0.1-mM EDTA, and 0.5% Triton X 100, pH 7.5) [91] and centrifuged at 1000× *g* for 10 min. The supernatant was collected, and the level of ROS was measured using a commercially available colorimetrical kit (#:KTE71621; Abbkine, Redlands, CA, USA), according to the manufacturer’s instructions.

### 4.9. Statistical Analyses 

All values were expressed as the mean ± SEMs. Any other experimental data, except for the pathological findings, were analyzed by one-way analysis of variance (ANOVA) followed by Duncan’s multiple range test. The injury scores of the liver pathological examination were analyzed using a *t*-test. A *p*-value < 0.05 was considered statistically significant.

## 5. Conclusions

The study demonstrated that fucoidan was effective in preventing TAA-induced hepatic damage in mice. The protective effect of fucoidan was due to its anti-inflammatory, antioxidant, and free radical scavenger effects. The underlying mechanisms included an increase in the expression of catalase, GPx, and SOD and a decrease in ROS production. The reduced inflammatory conditions were revealed by a decrease in the TNF-ɑ, IL-1β, and CRP levels and the inhibition of hepatic IL-6 and PNPLA3 expression, as well as an under-expression of the L-FABP, COX-2, and iNOS mRNA levels, which ultimately prevented the progression of hepatic damage. Hence, this study not only validated the use of fucoidan as a hepatoprotective agent but also supported its future exploration as a therapeutic drug for liver disorders.

## Figures and Tables

**Figure 1 molecules-26-01937-f001:**
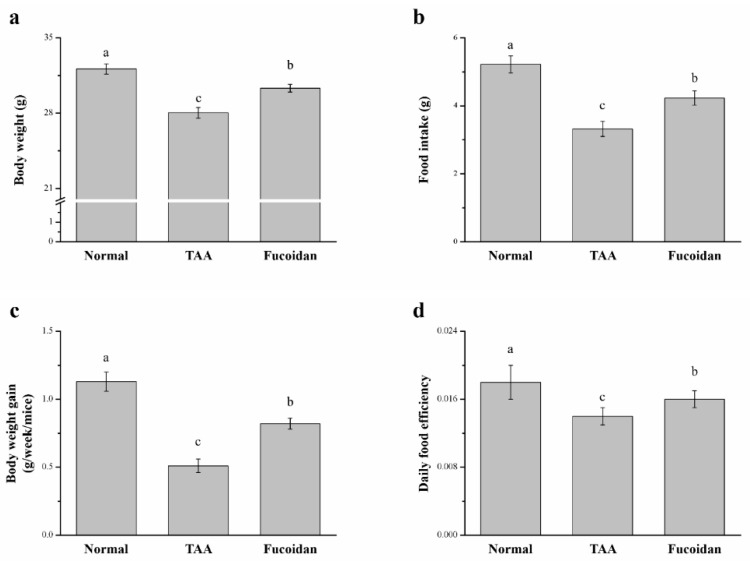
Changes of the (**a**) body weight, (**b**) food intake, (**c**) weekly body weight gain, and (**d**) daily food efficiency in normal mice, mice with thioacetamide (TAA)-induced liver injury, and fucoidan-treated mice with TAA-induced liver injury within a time period of 42 days. All data are presented as the means ± standard error (SEMs) (*n* = 10). ^a–c^ Data with different letters in the columns are significantly different with one-way analysis of variance (ANOVA), and the means of different groups were compared by Duncan’s test at *p* < 0.05.

**Figure 2 molecules-26-01937-f002:**
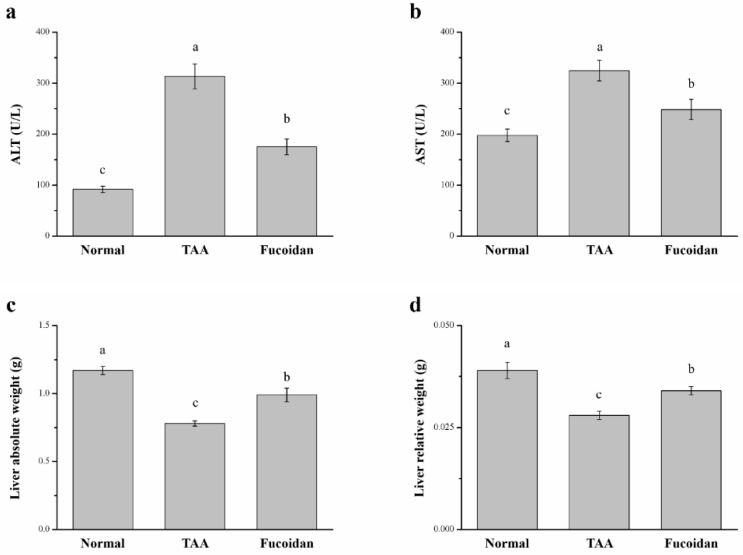
Changes in the (**a**) serum alanine aminotransferase (ALT) levels, (**b**) serum aspartate aminotransferase (AST) levels, (**c**) absolute weights, and (**d**) body weight-normalized weights (%) of the livers in normal mice, mice with TAA-induced liver injury, and fucoidan-treated mice with TAA-induced liver injury over a period of 42 days. All data are presented as the means ± SEMs (*n* = 10). ^a–c^ Data with different letters in the columns are significantly different with one-way ANOVA, and the means of different groups were compared by Duncan’s test at *p* < 0.05.

**Figure 3 molecules-26-01937-f003:**
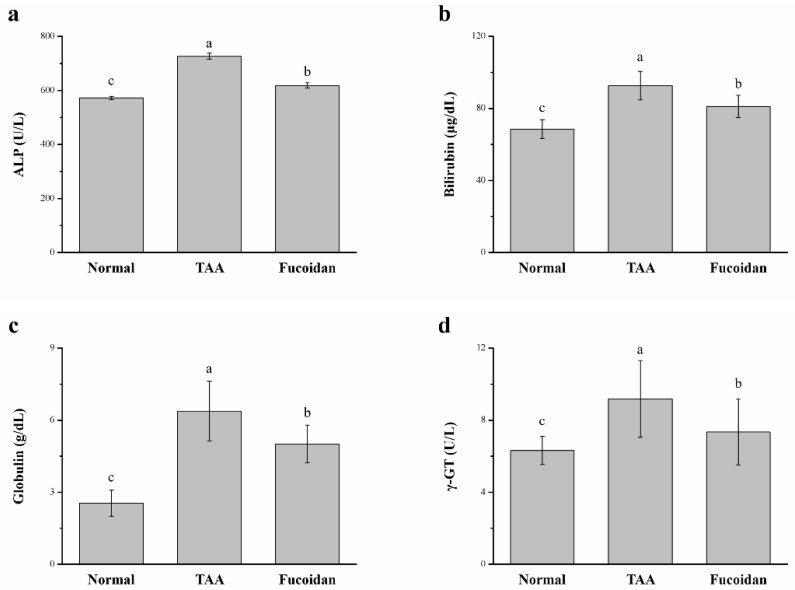
Changes of the serum (**a**) alkaline phosphatase (ALP), (**b**) bilirubin, (**c**) globulin, and (**d**) γ-glutamyl transferase (γ-GT) levels in normal mice, in mice with TAA-induced liver injury, and fucoidan-treated mice with TAA-induced liver injury over a period of 42 days. All data are presented as the means ± SEMs (*n* = 10). ^a–c^ Data with different letters in the columns are significantly different with one-way ANOVA, and the means of the different groups were compared by Duncan’s test at *p* < 0.05.

**Figure 4 molecules-26-01937-f004:**
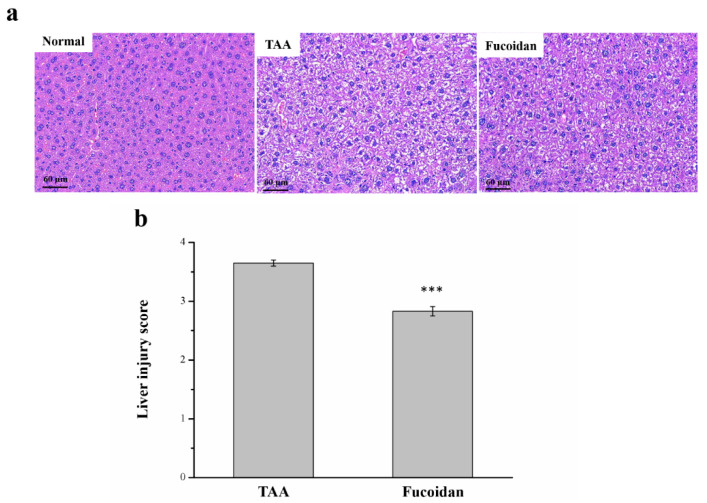
Changes of the (**a**) liver histology by hematoxylin and eosin (H&E) staining (magnification 200×), and (**b**) the scoring index of liver injuries in normal mice, mice with TAA-induced liver injury, and fucoidan-treated mice with TAA-induced liver injury over a period of 42 days. All data are presented as the means ± SEMs (*n* = 10). Statistical significance at *** *p* < 0.001.

**Figure 5 molecules-26-01937-f005:**
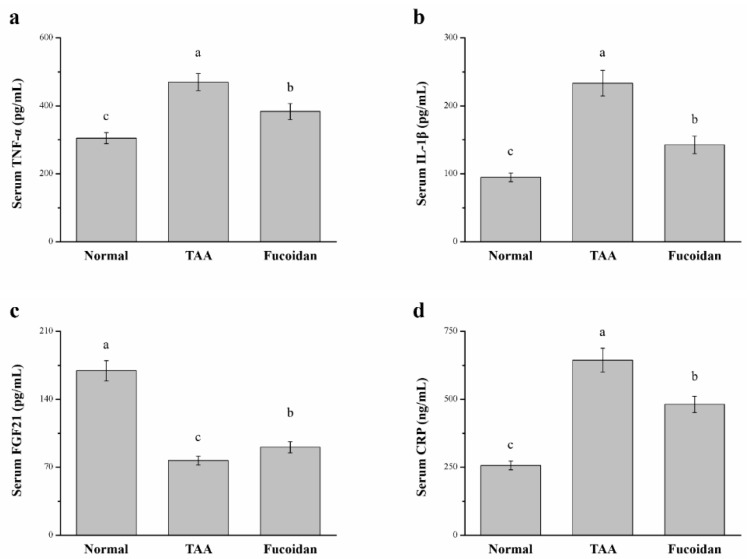
Changes of the serum (**a**) tumor necrosis factor-α (TNF-α), (**b**) interleukin-1β (IL-1β), (**c**) fibroblast growth factor-21 (FGF21), and (**d**) C-reactive protein (CRP) levels in normal mice, mice with TAA-induced liver injury, and fucoidan-treated mice with TAA-induced liver injury after 42 days. All data are presented as the means ± SEMs (*n* = 10). ^a–c^ Data with different letters in the columns are significantly different with one-way ANOVA, and the means of different groups were compared by Duncan’s test at *p* < 0.05.

**Figure 6 molecules-26-01937-f006:**
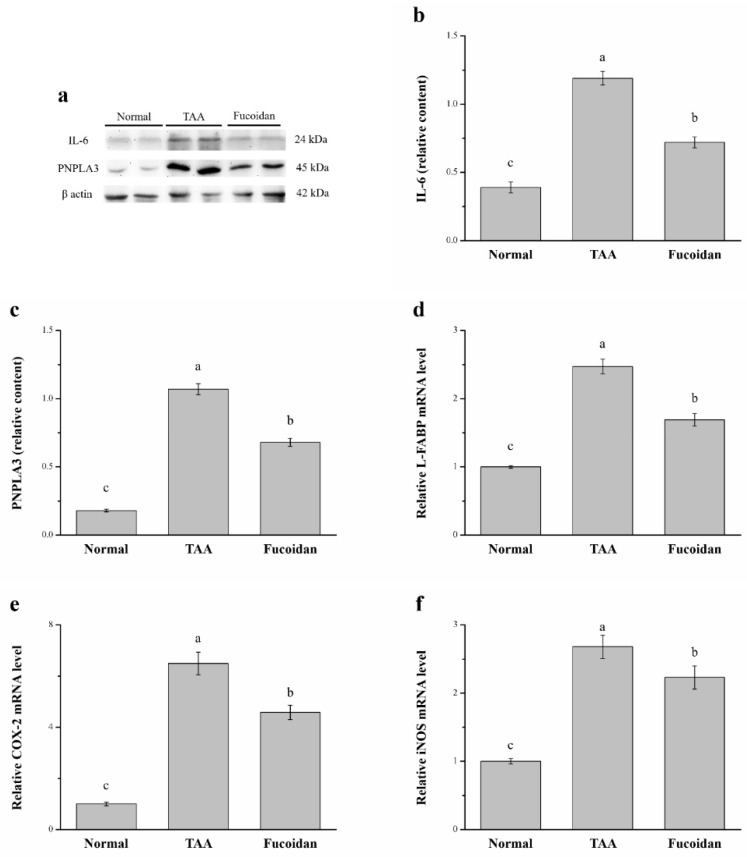
(**a**) Representative Western blot of the liver extracts for interleukin-6 (IL-6) and patatin-like phospholipid domain containing protein 3 (PNPLA3) expression, (**b**) IL-6, and (**c**) PNPLA3 expression, as well as changes in the (**d**) liver fatty acid-binding protein (L-FABP), (**e**) cyclooxygenase-2 (COX-2), and (**f**) inducible nitric oxide synthase (iNOS) mRNA levels in the livers of normal mice, TAA-induced liver injury mice, and fucoidan-treated TAA-induced liver injury mice over a time period of 42 days. All data are presented as the means ± SEMs (*n* = 10). ^a–c^ Data with different letters in the columns are significantly different with one-way ANOVA, and the means of different groups were compared by Duncan’s test at *p* < 0.05.

**Figure 7 molecules-26-01937-f007:**
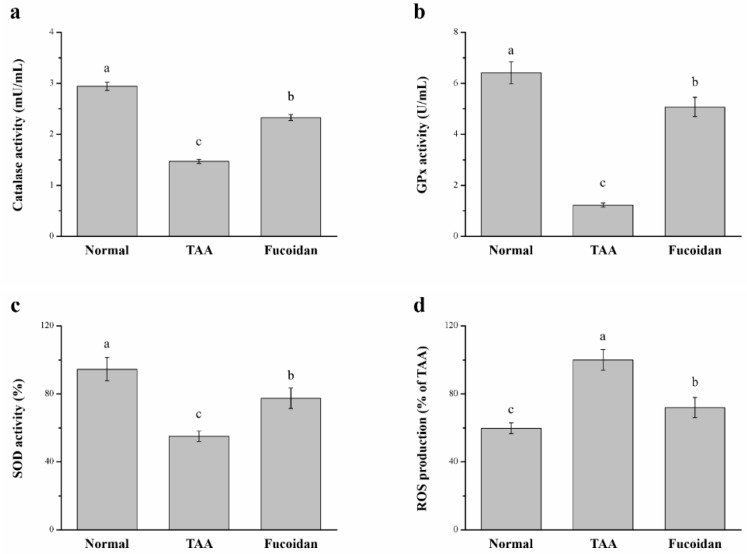
Changes of the hepatic (**a**) catalase activity, (**b**) glutathione peroxidase (GPx) activity, (**c**) superoxide dismutase (SOD) activity, and (**d**) reactive oxygen species (ROS) levels in normal mice, mice with TAA-induced liver injury, and fucoidan-treated mice with TAA-induced liver injury over a period of 42 days. All data are presented as the means ± SEMs (*n* = 10). ^a–c^ Data with different letters in the columns are significantly different with one-way ANOVA, and the means of different groups were compared by Duncan’s test at *p* < 0.05.

## Data Availability

The data presented in this study are available on request from the corresponding author.

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
