# Peer review of "The Ameliorative Effects of Fucoidan in Thioacetaide-Induced Liver Injury in Mice"

_molecules, 2021, doi:10.3390/molecules26071937_

Round 1

Reviewer 1 Report

Revision 09.03.2020

Molecules-1145258: ,,The Ameliorative Effects of Fucoidan in Thioacetamide-in- Duced Liver Injury in Mice’’

The article addresses the problem of cancer and liver disease research using fucoidan. Written in a clear, understandable way, accessible to any reader. The animals (mice) used for the research were cared according to the norms and instructions. The article uses 90 recent literature items.

Some minor remarks for improvement are listed below.

  1. The introduction talks about reducing cancer cells in mice. Where do mice with cancer or mice with liver damage come from? Can these cancer cells be transplanted? So, cancerous ones are transplanted in place of healthy ones?
  2. Werse 124-132, Does this mean that healthy mice would also gain weight after consuming turmeric or fucoidan? Can it be the same with people?
  3. Werse 12-121:..mice exhibited upregulated parameters in morphology. Question:
  4. Were parameters other than AIAT and Astat tested? Such typical blood tests of ESR type, morphology, are they important? If not why not?
  5. Werse 164: How was this liver damaged?
  6. Before discussing the results, the article should include items such as: Raw materials and research methods (eg. SEM and others).
  7. Werse 186: using the Suzuki scoring – 6. It would be necessary to expand this concept, perhaps it is known in the medical industry, but someone who would like to read about oxidants, about ways to "help" his liver, might not know this term. Similarly, Duncan's test or Sirius red staining. This should be described in the Methodology.
  8. The "Materials and Methods" section should be at the beginning of the article.
  9. I think that abbreviations like AFT, GPx, SOD and others should also be expanded when they are used for the first time in the text.
  10. Werse 441 ,,… certified veterinary pathologists blinded to the animals’ treatments.’’ The word "blinded" - what does it mean here?
  11. Werse 483 ,,…in ice cold lysis solution’’, What means ,,lysis’’.

Reviewer 2 Report

Fucoidan is a sulfated heteropolysaccharide found in brown algae and some echinoderms. Numerous studies of the last 10-15 years have been devoted to the biological action of fucoidans. Thus, fucoidans exhibit an extremely wide range of biological activities, which is the reason for the increased interest in them. So, in the literature there are reports on antitumor, immunomodulatory, antibacterial, antiviral, anti-inflammatory and other properties of fucoidans. The authors carried out a full-scale study in linear mice that demonstrates that fucoidan is effective in preventing thioacetamide-induced liver damage in mice. A pronounced protective effect of fucoidan was shown due to its anti-inflammatory, antioxidant and acceptor action. As a result, within the framework of this work, not only confirms the use of fucoidan as a hepatoprotective agent, but also suggests its future studies as a therapeutic drug in liver diseases.

Certainly, this study can be published in the journal Molecules, after some minor edits.

  1. Considering that the main audience of the Molecules journal is specialists in the field of organic and medicinal chemistry, I think it will be justified to add the structure of fucoidan in one of the figures in the introductory part of the article.
  2. For greater clarity and improve the visual perception of the material, I propose to convert the data from table 1 into a bar chart similarly to those given throughout the text of the article.
  3. Please indicate how many mice were taken into the experiment, as well as how many individuals were in each control group?
  4. Was food refusal or decreased food intake observed in thioacetamide-treated mice?
  5. There are extra spaces and dashes in words in the text.
